# BoMM: Multi-Modality Large-Small Model Bidirectional Collaboration

## Abstract

Different from existing single-modality large-small model collaborations, multi-modality large-small model collaboration is an under-explored paradigm where cloud-side multi-modality large model (MM-LM) collaborates with parties' small models (SMs) to achieve bidirectional domain-specific performance improvements. Nevertheless, this paradigm faces two key challenges. First, MM-LM inherently relies on abundant modality-aligned samples for training, but geographical and device diversity across parties inevitably lead to different collected samples and modalities. These differences significantly reduce overlapping sample entities across parties' multi-modality datasets, creating **modality alignment scarcity** challenge. Second, collected device failure and human annotation costs further lead to different modality missing problems in each party's dataset. Existing modality completion methods typically require enough modality-completed training samples to ensure generation quality, creating a **modality completeness gap** challenge. To address these challenges, we propose a **m**ulti-**m**odality large-small model **b**idirectional c**o**llaboration framework, named **BoMM**, which consists of two key components. Specifically, *global prototype-guided alignment* strategy identifies potentially aligned samples through similarity distribution comparisons between unaligned data and established global prototypes, enabling knowledge transfer from SMs to MM-LM. With established prototypes, *preference-driven modality adaptive completion* method integrates direct preference optimization into generator training with real-time scheduling to dynamically complete missing modalities, enabling knowledge transfer from MM-LM to SMs. Theoretical analysis confirms BoMM's $O(1/\sqrt{T})$ convergence rate. Across three multi-modality scenarios, it outperforms state-of-the-art methods by up to 6.64% on two well-known datasets. Our code is available at https://anonymous.4open.science/r/MultiLM-5D65.

## 1 Introduction

In large language model (LLM) and small models (SMs) collaborative scenarios, cloud-side LLM leverages parties' domain data and SMs' expertise to address domain-specific weaknesses, while SMs of parties can also benefit from LLM' superior reasoning and generation capabilities Wang et al. (2024a); Liu et al. (2024). This collaborative relationship extends to federated learning scenarios Guo et al. (2024) when privacy becomes essential Zhang et al. (2025), enabling mutual enhancement while keeping all raw data local to satisfy privacy requirements.

Existing large-small model collaborations utilize three primary approaches: (i) distillation-based methods Zhou et al. (2021); Liu et al. (2021); Pham et al. (2021) leverage knowledge distillation to transfer knowledge between domain-specific SMs and LLM in both directions; (ii) generation-based methods Zhang et al. (2021); Li & Jin (2022); Nasser et al. (2024) leverage synthetic datasets to transfer knowledge from SMs to LLMs or use LLMs' generative capabilities to create task-specific datasets; and (iii) parameter-based methods Fan et al. (2024); Cheng et al. (2021); Yu et al. (2023) employ parameter-efficient fine-tuning (PEFT) techniques Han et al. (2024) to transmit knowledge bidirectionally or selectively transfer parametric knowledge from LLMs to SMs.

Despite significant achievements, these methods focus on single-modality large-small model collaoration, neglecting multi-modality scenarios. In reality, with multi-modality data becoming more

prevalent, multi-modality large model (MM-LM) are becoming an increasingly important research direction Tong et al. (2025). Since MM-LM often exhibits suboptimal performance on domain-specific tasks Ye et al. (2024), they require access to multi-modality domain data from parties. This necessitates collaboration between multi-modality large and small models. Although a recent study Chen et al. (2025) aligns repre-

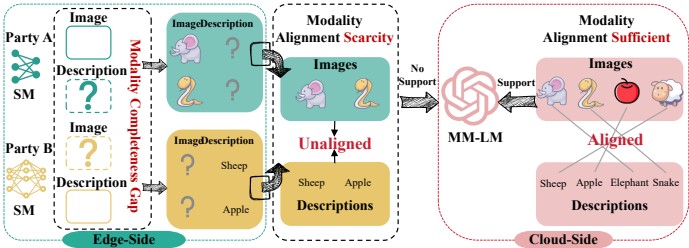

Figure 1: Multi-modality large-small model collaboration face to key challenges: modality completeness gaps from parties' modality missing, and modality alignment scarcity from parties' modality alignment problems for the same sample, hindering MM-LM training that relies on abundant modality alignments.

sentations between single-modality MM-LM and multi-modality SMs in medicine, it relies heavily on two idealistic assumptions: modality alignment and modality completeness for each sample. These rarely hold in real-world distributed settings. Additionally, this work also overlooks the need for mutual performance improvements in multi-modality collaborations, where the limitation of mutual performance improvement is also observed in single-modality collaborations.

Therefore, we explore an under-explored paradigm: multi-modality large-small model bidirectional collaboration, as shown in Fig. 1. Different from existing large-small model collaborations, due to geographical and device diversity across parties, this paradigm faces a **modality alignment scarcity** challenge where parties possess multi-modal data but lack aligned samples across modalities. These aligned samples are essential for training MM-LM's alignment capability Wu et al. (2023); Song et al. (2023). Moreover, we identify an additional critical challenge: **modality completeness gap**, which occurs when parties have incomplete or missing modalities due to device failures or annotation expenses. Existing completion approaches Wu et al. (2024a) still struggle to generate high-quality modalities with severe or even complete modality missing in training datasets.

To address these challenges, we propose a *m*ulti-*m*odality large-small model **b**idirectional **c**ollaboration framework, denoted as **BoMM**, where a *global prototype-guided alignment* method identifies potentially aligned samples by comparing similarity distributions to established global prototypes, enabling knowledge transfer from SMs to MM-LM. Building on these prototypes, *preference-driven modality adaptive completion* method optimizes the global generator to dynamically fill local missing modalities with a real-time sample scheduler, facilitating effective knowledge transfer from MM-LM to SMs. The main contributions are summarized as follows:

(a) We present an under-explored paradigm: **m**ulti-**m**odality large-small model **b**idirectional **c**ollaboration, and propose BoMM, where cloud-side MM-LM enhances domain performance by SMs, while parties' SMs overcome modality missing using MM-LM's generation capability.

(b) We mitigate modality alignment scarcity challenge through a global prototype-guided alignment method, and address modality completeness gap challenge via a preference-driven modality adaptive completion method, where a real-time sample scheduler is designed to filter generated modalities, enabling effective bidirectional knowledge transfer between models.

(c) BoMM can effectively handle different modality alignment situations, while supporting various modality missing rates of parties. Our theoretical analysis guarantees convergence at $O(1/\sqrt{T})$ rate, ensuring robust performance in multi-modality large-small model collaborations.

(d) Extensive experiments on two real-world multi-modal datasets across three scenarios clearly demonstrate that BoMM achieves substantial improvements over state-of-the-art methods under varying rates of aligned samples and modality missing.

## 2 RELATED WORK

### 2.1 LARGE-SMALL MODEL COLLABORATION

large-small model collaboration leverages distributed private domain data to address public data scarcity and domain-specific poor performance challenges in large language models (LLMs). Con-

currently, the extensive knowledge within LLMs can improve complex reasoning capabilities and provide comprehensive background information for domain-specific small models (SMs). Three primary methods facilitate knowledge transfer between SMs and LLMs Liu et al. (2025): distillation-based Zhou et al. (2021); Liu et al. (2021); Pham et al. (2021), generation-based Zhang et al. (2021); Li & Jin (2022); Nasser et al. (2024), and parameter-based transfer methods Fan et al. (2024); Cheng et al. (2021); Yu et al. (2023). These approaches primarily concentrate on unidirectional knowledge transfer between large-small models, with few works Fan et al. (2024); Cheng et al. (2021) exploring bidirectional knowledge exchange. More importantly, existing works fail to cope with collaborative scenarios for a multi-modality large model (MM-LM) and SMs, where parties may possess heterogeneous data modalities, diverse model architectures, and face varying degrees of modality incompleteness in their local datasets.

## 2.2 LARGE MULTI-MODALITY MODEL

The rapid advancement of multi-modality large model has significantly demonstrated their remarkable ability to tackle various multi-modality challenges in real-world applications, including personal AI assistants Wang et al. (2024b), education Lee et al. (2025), medicine Wang et al. (2024c), autonomous driving Cui et al. (2024), and others. However, these works depend on training data where multiple modalities (such as images and text) are paired for each sample, a condition rarely met in distributed environments. In these distributed environments, each party typically holds only specific modalities for their local samples, while complementary modalities for these exact samples are often entirely absent from other parties' collections. This modality misalignment generally emerges from regional differences, domain specializations, and other organizational factors, making complete multi-modality pairing exceedingly difficult to achieve. Leveraging such unaligned multi-modality data for MM-LM training can lead to weakened multi-modality understanding, increased hallucinations, and unreliable correlations between different modalities, ultimately limiting the model's practical utility in complex real-world scenarios Song et al. (2023).

## 3 PRELIMINARY

### 3.1 MULTI-MODALITY LARGE-SMALL MODEL COLLABORATION

In our multi-modality large-small model collaboration scenarios with $K$ parties and modality set $M$, each party $k \in \{1, 2, ..., K\}$ has dataset $D_k$ containing $N_k$ samples and $M_k \subseteq M$ modalities. Parties may have identical or different modalities due to modality missing, with different sample alignment ratios. Within dataset $D_k$, each party has labeled samples $(x_i^l, y_i^l) \in D_k^l$ and unlabeled samples $(x_i^{ul}, y_i^{ul}) \in D_k^{ul}$. These data include aligned samples $(x_i^a, y_i^a) \in D_k^a$ (same sample ID across $K$ parties) and unaligned samples $(x_i^{ua}, y_i^{ua}) \in D_k^{ua}$ (sample ID found only in party $k$).

### 3.2 OPTIMIZATION OBJECTIVE

Resource limitations lead parties to deploy lightweight small models (SMs) locally while collaboratively training a powerful domain-specialized multi-modality large model (MM-LM) on the cloud. Our framework establishes bi-directional knowledge transfer, simultaneously incorporating domain-specific expertise from SMs into MM-LM while enabling MM-LM to enhance SMs' performance. Formally, we denote MM-LM with parameters $\psi$ as $f_\psi$ and SM of party $k$ with parameters $\theta_k$ as $f_{\theta_k}$. Our bi-directional optimization objective includes:

**Global MM-LM**: The cloud-side multi-modality large model $f_\psi$ is optimized using parties' multi-modality data for both domain-specific tasks and modality completion:

$$\min_\psi \mathcal{L}_{\text{global}}(\psi; \{D_k\}_{k=1}^K, \{\theta_k\}_{k=1}^K), \tag{1}$$

where $\psi = [\psi_1, \psi_2]$, with $\psi_1$ and $\psi_2$ representing classifier and generator parameters, respectively.

**Local SMs**: Each party's small model $f_{\theta_k}$ is optimized using its local dataset $D_k$ for domain-specific tasks, improved by enhanced domain data from MM-LM:

$$\min_{\theta_k} \mathcal{L}_{\text{local}}(\theta_k; D_k, \psi), \tag{2}$$

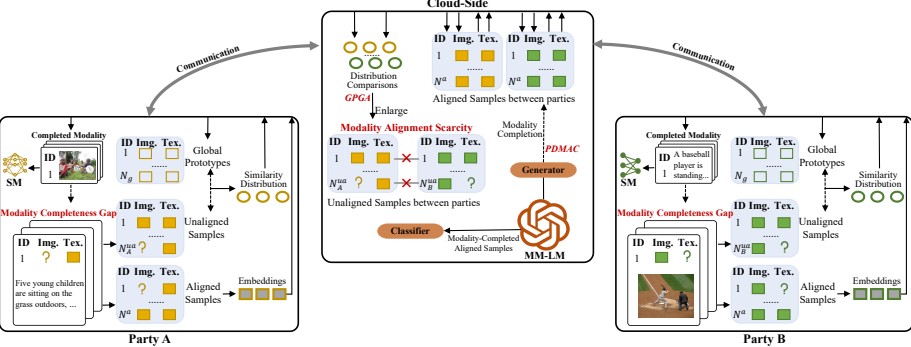

Figure 2: The overview of BoMM framework. The global prototype-guided alignment (GPGA) method identifies essential modality-aligned samples by analyzing similarity distributions between unaligned data and global prototypes, enhancing MM-LM's domain performance. Meanwhile, parties leverage a preference-driven modality adaptive completion (PDMAC) method to synthesize missing modality using MM-LM's generation ability, improving local domain performance.

where $\mathcal{L}_{local}$ denotes the party-specific optimization loss for each local model, and $\mathcal{L}_{global}$ represents the optimization loss for global MM-LM in the cloud.

## 4 OUR METHODOLOGY

In this section, we propose BoMM, a multi-modality large-small model bidirectional collaboration framework including a preference-driven modality adaptive completion method with a real-time sample scheduler and a global prototype-guided alignment strategy. The overall architecture is illustrated in Fig. 2.

### 4.1 PREFERENCE-DRIVEN MODALITY ADAPTIVE COMPLETION

The modality incompleteness of each party degrades local model performance on private domain tasks while simultaneously limiting knowledge contribution to cloud-side MM-LM training. Resource constraints prevent parties from training domain-specific MM-LM locally for modality completion. While general MM-LM is accessible via APIs, they generate generic outputs that fail to synthesize missing modalities for specialized domains accurately.

#### 4.1.1 MODALITY ADAPTIVE COMPLETION

Inspired by direct preference optimization (DPO) Rafailov et al. (2023), which guides models to generate human-preferred outputs, we design a preference-driven modality adaptive completion (PDMAC), enabling MM-LM to generate domain-specific modality information tailored to local private data. Unlike existing DPO approaches Wu et al. (2024b); Karthik et al. (2024) for modality completion that require extensive modality-completed training datasets, our method effectively handles different modality incompleteness or entire modality missing. By leveraging MM-LM for modality completion, we substantially improve model performance on domain-specific tasks.

To be specific, for $i$-th sample with existing modality information $x_{i,k}^{\{v,t\}}$, we create a preference pair $(g_i^p, g_i^q)$. $g_i^p$ is the preferred output, which is obtained from the closest global prototype $o_i$ to the $i$-th sample, while $g_i^q$ is the non-preferred output, which is the missing modality generated by the MM-LM's generator $\mathcal{G}$. The optimization objective for $\mathcal{G}$ is:

$$\mathcal{L}_{\text{prefer}} = -\mathbb{E}_{(x_{i,k}^{\{v,t\}}, g_i^p, g_i^q) \sim D_k} \left[ \log \sigma(\beta(R(x_{i,k}^{\{v,t\}}, g_i^p) - R(x_{i,k}^{\{v,t\}}, g_i^q)) \right], \tag{3}$$

where $\{v,t\}$ indicates available image ($v$), text ($t$), or both modalities. $\sigma$ is the logistic function, and $\beta$ scales the reward function $R(\cdot)$. Since parties may have different missing modalities, $R(\cdot)$ is adapted based on the generated modality type, following Jiang et al. (2024) for text and Karthik et al. (2024) for images.

Additionally, to prevent generated modality from lacking personalization and becoming too similar to its global prototype, we introduce an additional loss term:

$$\mathcal{L}_{\text{special}} = \|g_i^q - g_i^p\|^2 (1 - \text{sim}(x_{i,k}^{\{v,t\}}, o_i^{\{v,t\}})), \tag{4}$$

where $o_i$ and $x_{i,k}^{v,t}$ share modalities $i$-th sample already has, while $g_i^q$ and $g_i^p$ represent the target modality to generate. Our method needs just one complete-modality sample per class as a prototype, which is practical to obtain. If unavailable initially, existing generation methods Kwon et al. (2024); Laina et al. (2019) still can initialize these prototypes. Once established, samples grouped around these prototypes progressively receive high-quality completion results during training. For samples already obtaining the target modality, these real data serves directly as preferred output $g_i^p$. The generator $\mathcal{G}$ is updated by $\mathcal{L}_{\mathcal{G}} = \mathcal{L}_{\text{prefer}} + \gamma \mathcal{L}_{\text{special}}$, where $\gamma$ is a balancing hyperparameter. The MM-LM classification task loss term $\mathcal{L}_{task}$ is calculated by the standard cross-entropy loss Zhang & Sabuncu (2018) with completed modality data.

### 4.1.2 DYNAMIC SAMPLE SCHEDULER

Parties use the cloud-side MM-LM to generate missing modalities and expand local datasets. To handle potentially noisy generated data, we implement a scheduler that dynamic select high-reward samples in each batch based on sample reward. Low-quality samples are kept in the training set, with their current generations continuing to serve as non-preferred outputs for further optimization.

**Reward definition.** Since task loss can be applied to measure sample value Huang et al. (2025), we design a reward function that dynamically evaluates generator improvement after each batch update. Our reward function $r(\cdot)$ combines sample-level and batch-level assessments, with batch-level reward measuring the reduction in average generated loss before and after training. In practice, we use batch-average generated loss as an approximation of the expected generated loss across the entire data distribution. For the $i$-th sample in batch $b$, the reward is:

$$r(a_k^{i,b}, \theta_{\mathcal{C}_k}^{b-1}, \theta_{\mathcal{C}_k}^b) = \underbrace{-\log \mathcal{P}_{\theta_{\mathcal{C}_k}^{b-1}}(y_k^i | \bar{x}_k^{i,b}) + \delta \mathcal{H}}_{\text{sample-level}} + \underbrace{\frac{\overbrace{\sum_{i=1}^{N_b} e^{\mathcal{L}_{\mathcal{G}}(a_k^{i,b}; \theta_{\mathcal{C}_k}^{b-1})}}^{A} - \overbrace{\sum_{i=1}^{N_b} e^{\mathcal{L}_{\mathcal{G}}(a_k^{i,b+1}; \theta_{\mathcal{C}_k}^b)}}^{Q}}{\max\left(\sum_{i=1}^{N_b} e^{\mathcal{L}_{\mathcal{G}}(a_k^{i,b}; \theta_{\mathcal{C}_k}^{b-1})}, \sum_{i=1}^{N_b} e^{\mathcal{L}_{\mathcal{G}}(a_k^{i,b+1}; \theta_{\mathcal{C}_k}^b)}\right)}}_{\text{batch-level}}, \tag{5}$$

where $a_k^{i,b} = (\bar{x}_k^{i,b}, y_k^i)$. $\theta_{\mathcal{C}_k}^{b-1}$ represents classifier parameters in batch $b$, and $\mathcal{P}_{\theta_{\mathcal{C}_k}^{b-1}}$ is the model's output probability distribution given input $a_k^{i,b}$. $N_b$ is the batch size. Term $A$ evaluates $\theta_{\mathcal{C}_g}^{b-1}$ on batch $b-1$, while term $Q$ evaluates $\theta_{\mathcal{C}_g}^{b+1}$ on batch $b+1$. $\bar{x}_k^{i,b}$ combines intermediate representations across all layers to utilize the model's full representational capacity.

**Scheduler design.** The rewards of generated samples serve as supervisory signals for training the dynamic scheduler $\mathcal{R}$. This scheduling process addresses the exploration-exploitation trade-off Berger-Tal et al. (2014), requiring a balance between discovering new sample spaces and utilizing known high-quality examples. We implement two networks after an encoder layer: $R^p$ for exploitation and $R^q$ for exploration.

Specifically, the exploit network $\mathcal{R}^p$ predicts sample rewards by mapping inputs to observed rewards, while the explore network $\mathcal{R}^q$ estimates uncertainty and adds an exploration bonus. This design balances exploitation and exploration during sample selection, following UCB Wen et al. (2015) and Thompson Sampling principles Zhang et al. (2020). For modality incomplete data $\bar{x}_k^{i,b} \in D_k^{\text{inc}}$ in $b$-th batch, $\mathcal{R}^p$ eedforward neural network with residual connections, denoted by $\mathcal{R}^p(\bar{x}_k^{i,b}; \theta_p^b)$. After obtaining the observed reward $r(a_k^{i,b}, \theta_p^{b-1}, \theta_p^b)$, the parameters $\theta_p^b$ are updated by:

$$\mathcal{L}^p(D_{k,b}^{\text{inc}}, \theta_p^b) = \frac{1}{2N_b} \sum_{i=1}^{N_b} \left[ \mathcal{R}^p(\bar{x}_k^{i,b}; \theta_p^b) - r(a_k^{i,b}, \theta_p^{b-1}, \theta_p^b) \right]^2. \tag{6}$$

The explore network $\mathcal{R}^q$ takes input $h_{i,b,k}^p$ created by concatenating intermediate hidden states of $\mathcal{R}^p(\bar{x}_k^{i,b}; \theta_p^{b-1})$ with the last dimension. This allows $\mathcal{R}^q$ to consider the exploit network's internal

states. Like $\mathcal{R}^p$, $\mathcal{R}^q$ is a feedforward neural network with residual connections. Its parameters $\theta q^b$ are updated by:

$$\mathcal{L}^q(D_{k,b}^{\text{inc}}, \theta_q^b) = \frac{1}{2N_b} \sum_{i=1}^{N_b} \left[ \mathcal{R}^q(h_{i,b,k}^p; \theta_q^b) - \left( r(a_k^{i,b}, \theta_p^{b-1}, \theta_p^b) - \mathcal{R}^p(\bar{x}_k^{i,b}; \theta_p^b) \right) \right]^2. \tag{7}$$

The final reward estimation combines both networks:

$$\hat{r}(\bar{x}_k^{i,b}; \theta_p^b, \theta_q^b) = \mathcal{R}^p(\bar{x}_k^{i,b}; \theta_p^b) + \lambda \mathcal{R}^q(h_{i,b,k}^p; \theta_q^b), \tag{8}$$

where $\lambda$ controls exploration strength. Samples with predicted rewards $\hat{r} > \mu$ are selected as high-value samples for the current batch, which $\mu$ is a hyperparameter.

## 4.2 GLOBAL PROTOTYPE-GUIDED ALIGNMENT

Finding aligned data across parties is difficult due to geographic and domain differences. This creates a significant challenge for MM-LM training, which requires aligned multi-modality data. Current alignment techniques Feng (2022); Feng et al. (2022) typically need some pre-existing aligned data and can't handle scenarios with variable alignment quantities, especially with zero-aligned samples. While Guo et al. (2025) addresses zero-aligned scenarios, it assumes uniform modality distribution across parties and complete modalities within samples, failing in multi-modality collaboration settings. To address these limitations, we propose a global prototype-guided alignment (GPGA) strategy that works with any number of pre-existing aligned data, including zero. We use optimal transport (OT) distance to derive relationship vectors between unaligned samples and global prototypes, then identify new aligned data by analyzing these relationship vectors.

Specifically, even with zero pre-existing aligned data, we start by using local labeled data $D_k^l$ to create class prototypes $\{p_k^c\}_{c=1}^C$ through weighted averaging Bien & Tibshirani (2011). We pair same-class prototypes across parties as initial aligned data to seed the global prototype set $\mathcal{O} = \{o^m | m = 1, ..., N_{\text{global}}\}$. Using optimal transport Peyré et al. (2019), we calculate transport cost $M_{im}^k = 1 - \cos(x_i^{ua}, o^m)$ between unaligned samples and global prototypes. For each unaligned sample, we create a cost vector $\mathbf{M}_i^k$ representing its relationship to all global prototypes. To find corresponding samples across parties, we compute:

$$j^\star = \text{argmax}_{j \in N_w^{ua}} \text{sim}(x_k^{ua,i}, x_w^{ua,j}) = \text{argmax}_{j \in N_w^{ua}} \frac{\mathbf{M}_i^k \cdot \mathbf{M}_j^w}{\|\mathbf{M}_i^k\|_2 \cdot \|\mathbf{M}_j^w\|_2}, \tag{9}$$

where $N_w^{ua}$ is the size of the unaligned dataset $D_w^{ua}$ in party $w$. This identifies cross-party alignment by comparing relationship patterns to shared prototypes rather than directly comparing incompatible modalities. The newly identified aligned sample pairs join the global prototype set until reaching $N_{\text{global}}$, after which these prototypes guide alignment of remaining samples. These enlarged aligned samples are utilized to improve the domain performance of local SM $\mathcal{L}_{task}^k$ by standard cross-entropy loss Zhang & Sabuncu (2018). These components support each other cyclically: alignment relies on completion for consistent representations, while completion depends on the global prototypes from alignment.

## 4.3 COMPLEXITY AND CONVERGENCE ANALYSIS

We analyze the convergence of our proposed BoMM. Let $\Psi := \{\psi\}$ and $\Theta := \{\theta_1, \theta_2, \cdots, \theta_{K-1}, \theta_K\}$. The optimization problems in Eq. 1 and Eq. 2 thus form a bilevel optimization problem, which can be formulated as follows.

$$\min_{\Psi} F(\Psi) := \mathcal{L}_{\text{global}}(\Psi, \Theta^*(\Psi)) = \mathbb{E}_\xi[L_{\text{global}}(\Psi, \Theta^*(\Psi); \xi)] = \frac{1}{n} \sum_{i=1}^n L_{\text{global}}(\Psi, \Theta^*(\Psi); \xi_i),$$

$$\text{s.t.} \quad \Theta^*(\Psi) = \arg\min_{\Theta} \mathcal{L}_{\text{local}}(\Psi, \Theta) = \mathbb{E}_\zeta[L_{\text{local}}(\Psi, \Theta; \zeta)] = \frac{1}{n} \sum_{i=1}^n L_{\text{local}}(\Psi, \Theta; \zeta_i),$$

$$\tag{10}$$

where $\mathcal{L}_{\text{global}}$ contains $\mathcal{L}_\mathcal{G}$ and $\mathcal{L}_{\text{task}}$. $\mathcal{L}_{\text{local}}$ consists only of $\mathcal{L}_{\text{task}}^k$. $L_{\text{global}}$ and $L_{\text{local}}$ are loss functions per sample for cloud and parties, where $L_{\text{local}}(\Psi, \Theta; \zeta) := l_{\text{local}}(\Psi, \Theta; \zeta) + \frac{\varphi}{2}\|\Theta\|_F^2$, with $\zeta$ representing a sample from $K$ parties. To analyze convergence, we introduce the following definition.

**Definition 1.** A point $\hat{\Psi}$ is called an $\epsilon$-*accurate stationary point* for the objective function $F(\Psi)$ if $\mathbb{E}\|\nabla F(\hat{\Psi})\|^2 \leq \epsilon$.

**Proof Sketch.** To explore essential insights, we first bound the tracking error $\|\Theta_t^{j-1} - \Theta^*(\Psi_t^0)\|$ between local parameters and optimal parameters at the $j$-th epoch. Then, using virtual updates technology (Yang et al., 2022), we establish an upper bound for the gradient approximation error $\left\|\frac{\partial \mathcal{L}_{\text{local}}(\Psi_t^j, \Theta_t^\tau)}{\partial \Psi_t^j} - \nabla F(\Psi_t^j)\right\|$.

**Theorem 1.** Under Assumptions 1-4 (detailed in Appendix B.1), define $\alpha := -L + \varphi$, choose step size $\eta$ to be $\frac{2}{L_2 + \alpha}$, $N_b = \mathcal{O}(\frac{1}{\sqrt{\epsilon}})$, $\tau = \mathcal{O}(\log \frac{1}{\epsilon})$, $\eta' < \frac{1}{2L_0}$ and suppose $\alpha < L_2$, we have:

$$\frac{1}{\tau'T} \sum_{t=0}^{T-1} \sum_{j=0}^{\tau'-1} \|\nabla F(\Psi_t^j)\|^2 \leq \frac{F(\Psi_0^0) - \inf_\Theta F(\Psi)}{\tau'T(\frac{\eta'}{2} - L_0\eta'^2)} + (\eta' + 2\eta'^2 L_0)L_2^2\Delta^2 \frac{\tau' - 1}{\tau'} + \mathcal{O}(\epsilon), \quad (11)$$

where $\mathcal{O}(\cdot)$ indicates growth rate proportional to or slower than $\epsilon$. With $L_0 := L_2 + \frac{2L_2^2 + L_1^2 L_3}{\alpha} + \frac{L_1 L_2 L_3 + L_1 L_2 L_4 + L_2^3}{\alpha^2} + \frac{L_1 L_2^2 L_4}{\alpha^3}$, setting $\tau' = 1$, yields a convergence rate of $\mathcal{O}(\frac{1}{\sqrt{T}})$.

## 5 EXPERIMENTS

### 5.1 EXPERIMENTAL SETUP

**Settings:** We evaluate our proposed BoMM framework on two widely-used multi-modal datasets: MS COCO Lin et al. (2014) and CUB-200-2011 Wah et al. (2011). Specifically, we conduct experiments in a large-small collaboration scenario with heterogeneous multi-modality models, including two parties' SMs and one cloud-side MM-LM. Each party possesses both aligned and unaligned samples, where aligned samples share identical IDs but contain different textual descriptions, while unaligned samples are exclusively owned by individual parties. We evaluate three modality missing scenarios. In scenario 1, both parties and MM-LM suffer from text modality missing with probability $\Delta$. In scenario 2, party 0 maintains complete image and text modalities while party 1 experiences text modality missing with probability $\Delta$ for both aligned and unaligned samples. In scenario 3, both parties and MM-LM suffer from image modality missing with probability $\Delta$. Details of experimental datasets, settings and implementation details are provided in Appendix A.

**Baselines:** We compare our BoMM framework against five baseline methods: **1) LOCAL**, which trains local models using only local data without cloud-side collaboration; **2) MM-LOCAL**, which employs local multi-modal models to complete missing modalities before local training; **3) Vanilla** aggregates aligned data representations at the cloud-side MM-LM while parties train on their local data; **4) MM-AC**, which extends Vanilla by using the MM-LM to complete missing modalities in aligned data; and **5) MM-UC**, that further incorporates modality completion for both aligned and unaligned data at the cloud side.

### 5.2 OVERALL PERFORMANCE

Tab. 1 presents the experimental results under both scenario 1 and scenario 2 with modality missing probability $\Delta$ set to 0.5. Our findings reveal that text modality completion provides limited performance gains for party's models, as evidenced by comparing MM-AC and MM-UC results where the inclusion of unaligned data completion shows minimal improvement. While this factor also constrains the improvement potential of our BoMM framework on local performance, our method still achieves consistent gains. Most notably, BoMM demonstrates substantial improvements on the cloud-side MM-LM performance. Specifically, under scenario 1 with zero aligned samples, BoMM achieves 4.75% improvement over the best baseline on MS COCO. Under scenario 2, the cloud-side improvement reaches 6.64% on MS COCO, highlighting the effectiveness of our bidirectional collaboration framework in enhancing domain-specific multi-modality large model capabilities.

Moreover, as shown in Tab. 2, scenario 3 demonstrates that image modality completion yields significantly greater performance improvements compared to text completion scenarios. This is evidenced by the substantial gains observed when comparing MM-UC against Vanilla, where Party 1's accuracy dramatically increases from 38.96% to 50.08%, representing an 11.12% improvement.

Table 1: Model performance comparison of BoMM under scenario 1 and scenario 2. P1 and P2 represent party 1 and party 2's performance, while C represents the cloud-side MM-LM performance.

| Aligned Number | Method | Scenario 1 | | | | | | Scenario 2 | | |
| | | MS COCO | | | CUB-200-2011 | | | MS COCO | | |
| | | P1 | P2 | C | P1 | P2 | C | P1 | P2 | C |
| --- | --- | --- | --- | --- | --- | --- | --- | --- | --- | --- |
| #0 | LOCAL | 38.67 | 38.28 | - | 56.34 | 29.98 | - | 50.48 | 34.96 | - |
| | MM-LOCAL | 36.13 | 36.52 | - | 57.62 | 31.84 | - | 50.68 | 36.52 | - |
| | Vanilla | 38.88 | 37.11 | 27.19 | 60.44 | 41.60 | 23.34 | 51.26 | 35.74 | 24.80 |
| | MM-AC | 39.06 | 37.40 | 27.64 | 59.45 | 46.38 | 23.27 | 50.20 | 35.84 | 26.26 |
| | MM-UC | 38.18 | 36.33 | 26.27 | 58.96 | 46.62 | 22.65 | 50.68 | 33.98 | 25.78 |
| | BoMM | 40.17 | 39.78 | 32.39 | 61.43 | 47.77 | 24.12 | 52.75 | 33.69 | 32.90 |
| | Imp. (%) | 1.99 | 2.38 | 4.75 | 0.99 | 1.15 | 0.78 | 0.49 | -2.15 | 6.64 |
| #200 | LOCAL | 39.47 | 37.41 | - | 60.85 | 37.25 | - | 50.98 | 37.41 | - |
| | MM-LOCAL | 39.14 | 36.10 | - | 60.36 | 35.58 | - | 51.97 | 36.10 | - |
| | Vanilla | 39.06 | 38.82 | 28.61 | 60.28 | 47.37 | 22.70 | 52.79 | 36.75 | 26.15 |
| | MM-AC | 38.15 | 39.80 | 28.45 | 60.03 | 43.25 | 23.11 | 51.94 | 36.84 | 25.65 |
| | MM-UC | 37.17 | 38.81 | 27.80 | 60.36 | 47.86 | 23.02 | 52.78 | 34.53 | 25.25 |
| | BoMM | 41.22 | 40.41 | 35.21 | 61.02 | 48.19 | 24.10 | 54.32 | 37.89 | 30.10 |
| | Imp. (%) | 1.75 | 0.61 | 6.60 | 0.17 | 0.82 | 0.99 | 1.53 | 1.05 | 3.95 |

Building upon this foundation, our proposed BoMM framework achieves even more pronounced improvements on local performance, with Party 1 and Party 2 achieving additional 2.27% and 2.28% accuracy gains respectively over the strongest baseline MM-UC. The cloud-side MM-LM also benefits significantly from this collaborative approach, improving from 44.90% to 51.18%, demonstrating a 6.28% enhancement. Furthermore, we evaluate the quality of generated missing modalities using CLIPScore metrics Hessel et al. (2021), where BoMM consistently outperforms all baselines with scores of 68.86 and 70.24 for both parties, indicating superior generation quality that contributes to the overall model performance improvements.

Table 2: Model performance under scenario 3.

| Method | MS COCO | | | | |
| | P1 | | P2 | | C |
| | Acc | CLIPScore | Acc | CLIPScore | Acc |
| --- | --- | --- | --- | --- | --- |
| LOCAL | 36.13 | - | 34.47 | - | - |
| MM-LOCAL | 30.18 | - | 36.62 | - | - |
| Vanilla | 38.96 | - | 38.18 | - | 27.83 |
| MM-AC | 40.79 | 64.12 | 39.88 | 65.67 | 28.04 |
| MM-UC | 50.08 | 62.16 | 39.72 | 67.56 | 44.90 |
| BoMM | 52.35 | 68.86 | 42.16 | 70.24 | 51.18 |
| Imp. (%) | 2.27 | 4.74 | 2.28 | 2.68 | 6.28 |

## 5.3 ABLATION STUDY

We conduct ablation studies on scenario 2 with zero aligned samples to validate the contribution of key components in our BoMM framework. Specifically, removing the preference loss corresponds to setting $\mathcal{L}_{\text{prefer}}$ to 0, while removing the special loss involves setting the

Table 3: Ablation study results.

| Model | MS COCO | | | CUB-200-2011 | | |
| | P1 | P2 | C | P1 | P2 | C |
| --- | --- | --- | --- | --- | --- | --- |
| BoMM | 40.17 | 39.78 | 32.39 | 61.43 | 47.77 | 24.12 |
| w/o preference loss | 37.51 | 38.13 | 30.82 | 60.48 | 45.18 | 20.34 |
| w/o special loss | 38.47 | 35.35 | 30.66 | 60.05 | 40.23 | 7.23 |
| w/o scheduler | 38.37 | 35.15 | 30.56 | 59.96 | 41.50 | 11.13 |

balancing hyperparameter $\gamma$ to zero. For the scheduler ablation, we set the reward threshold $\mu$ to 0, which disables the sample filtering mechanism and includes all generated samples in training. The results in Tab. 3 demonstrate that each component contributes meaningfully to the overall performance of BoMM. Removing these components both lead to consistent performance degradation across both parties and the cloud-side MM-LM.

## 5.4 CONVERGENCE EXPERIMENTS

Fig. 3(A) demonstrates the convergence behavior of our proposed BoMM framework compared with baseline methods across 5 training rounds. We evaluate convergence performance under the challenging zero-aligned samples scenario on both MS COCO and CUB-200-2011 datasets. Notably, BoMM exhibits faster convergence and reaches higher final accuracy compared to baseline methods, with both local parties and the MM-LM demonstrating steady performance improvements throughout the training process.

## 5.5 HYPERPARAMETER ANALYSIS

Fig. 3(B) presents the sensitivity analysis of two key hyperparameters in our BoMM framework. In Fig. 3(i), we evaluate the impact of modality missing rate $\Delta$ on performance across both datasets.

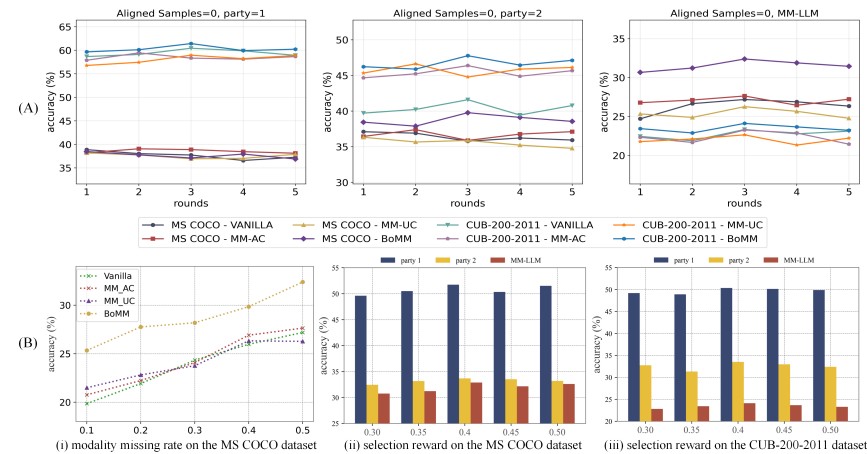

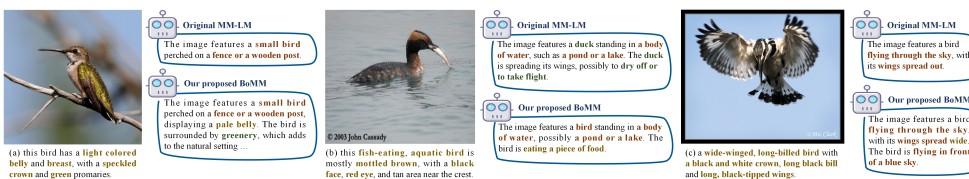

Figure 3: Convergence analysis and hyperparameter analysis of our proposed BoMM.



Figure 4: Case study results.

As expected, increasing modality missing rates lead to performance degradation, with our method maintaining stable performance even under severe missing conditions. Fig. 3(ii) and (iii) shows the effect of sample selection threshold $\mu$ on model performance. The results indicate that an optimal threshold around $\mu = 0.4$ achieves the best balance between sample quality and quantity, with too low thresholds including noisy samples and too high thresholds limiting data availability.

### 5.6 CASE STUDY

The case study results in Fig. 4 compares text descriptions generated by the original MM-LM against our proposed BoMM method across three representative bird images. The color coding in the descriptions indicates different types of attributes: yellow bold text represents attributes that accurately match the dataset's standard descriptions, red bold text indicates attributes generated by the models that don't align with the dataset descriptions, and green bold text highlights obviously incorrect attributes. Specifically, BoMM consistently produces more detailed and contextually rich descriptions. For example, while the original method provides a basic description of "a small bird perched on a fence" our BoMM method enhances this with additional details such as "displaying a pale belly" and "surrounded by greenery". These improvements demonstrate that our preference-driven modality adaptive completion method successfully generates domain-specific, high-quality descriptions that better capture visual details.

## 6 CONCLUSION AND FUTURE DIRECTION

This paper explores an under-explored paradigm: multi-modality large-small model collaboration. We propose a global prototype-guided alignment method to identify potentially aligned samples to mitigate modality alignment scarcity, and design a preference-driven modality adaptive completion method to address modality completeness gap challenges, enabling bidirectional large-small model knowledge transfer. Although we avoid direct leakage of raw data by exchanging intermediate representations, enhancing privacy remains an important future direction.

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

## A   IMPLEMENT DETAILS

We evaluate our proposed BoMM framework on two widely-used multi-modal datasets: MS COCO Lin et al. (2014) and CUB-200-2011 Wah et al. (2011). Specifically, the MS COCO dataset contains images across 80 object categories, where each image is paired with at least five textual descriptions. For our image-text classification task, we assign each image to the category with the highest number

of target objects present. The CUB-200-2011 dataset comprises 11,788 images covering 200 fine-grained bird species, with 5,994 images for training and 5,794 for testing. Following with Reed et al. (2016), we utilize the extended version that incorporates fine-grained visual descriptions, where each image contains at least 10 descriptions, with each description comprising at least 10 words without species names, background details, or action information.

We conduct experiments in a large-small collaboration scenario with heterogeneous multi-modality models, including two parties' SMs and one cloud-side MM-LM. The parties deploy small models including CLIP-base[1] and CLIP-large[2] locally, while the cloud hosts a large multi-modal model LLaVA-1.5-7B[3].Each party possesses both aligned and unaligned samples, where aligned samples share identical IDs but contain different textual descriptions, while unaligned samples are exclusively owned by individual parties. We evaluate three modality missing scenarios. In scenario 1, both parties and MM-LM suffer from text modality missing with probability $\Delta$. In scenario 2, party 0 maintains complete image and text modalities while party 1 experiences text modality missing with probability $\Delta$ for both aligned and unaligned samples. In scenario 3, both parties and MM-LM suffer from image modality missing with probability $\Delta$. In addition, to assess the impact of data alignment, we vary the number of aligned samples across 0 and 200, while fixing unaligned samples at 1,000 for each party and MM-LM.

We set the collaboration process to 3 global rounds, with a batch size of 32. In each round, both parties and the MM-LM perform 20 training epochs. All models are optimized using the Adam optimizer Kingma (2014), with learning rates set to 5e-4 for parties and 5e-5 for the MM-LM. Finally, we use accuracy to measure classification performance, CLIPScore Hessel et al. (2021) to evaluate the quality of generated modalities, communication cost measured in MB to assess data transfer efficiency, and computational time in minutes to evaluate system efficiency.

## B PROOF OF THEOREM 1

### B.1 ASSUMPTIONS

Let $\mathcal{Z} := \{\Psi, \Phi\}$ denote all model parameters. We adopt standard assumptions for bi-level optimization problem (Ghadimi & Wang, 2018; Ji et al., 2020) equation 10 on $L_{\text{global}}$ and $L_{\text{local}}$.

**Assumption 1. (Lipschitz Condition)** The loss function $L_{\text{global}}(\mathcal{Z}; \xi)$ is $L_1$-Lipschitz for any given $\xi$.

**Assumption 2. (Smoothness)** The loss functions $L_{\text{global}}(\mathcal{Z}; \xi)$, $L_{\text{local}}(\mathcal{Z}; \xi)$ and $l_{\text{global}}(\mathcal{Z}; \xi)$ are $L_2$-smooth, $L_2$-smooth and $L$-smooth for any give $\xi$ and $\zeta$, respectively.

**Assumption 3. (Lipschitz Condition for Second Derivatives)** The second derivatives $\nabla_\Psi \nabla_\Theta L_{\text{local}}(\mathcal{Z}; \zeta)$ and $\nabla_\Theta^2 L_{\text{local}}(\mathcal{Z}; \zeta)$ are $L_3$-Lipschitz and $L_4$-Lipschitz for any give $\zeta$, respectively.

**Assumption 4. (Bounded Domain)** The parameter $\Theta$ is in a bounded domain with a diameter $\Delta$, i.e., for any $\Theta_1$ and $\Theta_2$, we have:
$$\|\Theta_1 - \Theta_2\| \leq \Delta.$$

### B.2 ABSTRACTION OF OUR ALGORITHM

For clarity in convergence analysis, we write our algorithm as Algorithm 1.

### B.3 PROOF DETAILS

To give the convergence analysis of our algorithm, we need to give some useful lemmas first.

Firstly, with $L$-smoothness of $l_{\text{local}}(\Psi, \Theta; \zeta)$, we could give the strongly convexity property of $L_{\text{local}}(\Psi, \Theta; \zeta)$.

---

[1]https://huggingface.co/openai/clip-vit-base-patch32

[2]https://huggingface.co/openai/clip-vit-large-patch14

[3]https://huggingface.co/llava-hf/llava-1.5-7b-hf

---

**Algorithm 1** Algorithmic Abstraction of BoMM

---

**Input:** communication round $T$; learning rate: $\eta$ and $\eta'$; epoch: $\tau, \tau'$; initial parameters $\Theta_{-1}^{\tau}$ and $\Psi_{-1}^{\tau'}$.

**for** $t = 0, 1, \ldots, T - 1$ **do**
    Let $\Theta_t^0 = \Theta_{t-1}^{\tau}$ and $\Psi_t^0 = \Psi_{t-1}^{\tau'}$;
    **for** $j' = 1, 2, \ldots, \tau'$ **do**
        Select $S_{j-1}'$, a subset of each party's dataset $D_k$,
        update $\Psi_t^{j'} = \Psi_t^{j'-1} - \eta' \nabla_{\Psi} L_{\text{global}}(\Psi_t^{j'-1}, \Theta_{t-1}^{\tau}; S_{j-1}')$;
    **end for**
    **for** $j = 1, 2, \ldots, \tau$ **do**
        Update $\Theta_t^j = \Theta_t^{j-1} - \eta \nabla_{\Theta} L_{\text{local}}(\Psi_t^{\tau'}, \Theta_t^{j-1})$;
    **end for**
**end for**

---

**Lemma 1.** Under Assumption 2, suppose $\alpha := -L + \varphi > 0$, $L_{\text{local}}(\Psi, \Theta; \zeta)$ is $\alpha$-strongly convex w.r.t $\Theta$.

Based on Assumptions 1 and 2, we could give Lemma 2 directly.

**Lemma 2.** Under Assumption 1 and 2, the derivatives $\nabla L_{\text{global}}(\mathcal{Z}; \xi)$, $\nabla_{\Psi} \nabla_{\Theta} L_{\text{local}}(\mathcal{Z}; \zeta)$ and $\nabla_{\Theta}^2 L_{\text{local}}(\mathcal{Z}; \zeta)$ have bounded variances, i,e., for any $\mathcal{Z}$, we have

$$\mathbb{E}_{\xi} \| \nabla L_{\text{global}}(\mathcal{Z}; \xi) - \nabla \mathcal{L}_{\text{global}}(\mathcal{Z}) \|^2 \leq L_1^2, \tag{12}$$

$$\mathbb{E}_{\zeta} \| \nabla_{\Psi} \nabla_{\Theta} L_{\text{local}}(\mathcal{Z}; \zeta) - \nabla_{\Psi} \nabla_{\Theta} \mathcal{L}_{\text{local}}(\mathcal{Z}) \|^2 \leq L_2^2, \tag{13}$$

$$\mathbb{E}_{\zeta} \| \nabla_{\Theta}^2 L_{\text{local}}(\mathcal{Z}; \zeta) - \nabla_{\Theta}^2 \mathcal{L}_{\text{local}}(\mathcal{Z}) \|^2 \leq L_2^2. \tag{14}$$

We omit them since the proof of Lemma 1 and 2 is basic.

To show the smooth property of $F(\Psi)$, we introduce the following lemma which is proposed in Ghadimi & Wang (2018) firstly.

**Lemma 3. (Lemma 2.2 in Ghadimi & Wang (2018))** Under Assumptions 1, 2 and 3, $F(\Psi)$ is $L_0$-smooth where $L_0$ is given by

$$L_0 := L_2 + \frac{2L_2^2 + L_1^2 L_3}{\alpha} + \frac{L_1 L_2 L_3 + L_1 L_2 L_4 + L_2^3}{\alpha^2} + \frac{L_1 L_2^2 L_4}{\alpha^3}.$$

Tracking error $\mathbb{E} \| \Theta_t^{j-1} - \Theta^*(\Psi_t^0) \|$ is an important component in our convergence analysis. To give an upper bound on the tracking error, we utilized Lemma 9 in Ji et al. (2021). For simplicity, we assume equal batch sizes $N_b$ for both cloud and local data.

**Lemma 4. (Lemma 9 in Ji et al. (2021))** Under Assumptions 1 and 2, with stepsize $\eta$ to be $\frac{2}{L_2 + \alpha}$, we have

$$\mathbb{E} \| \Theta_t^{j-1} - \Theta^*(\Psi_t^0) \|^2 \leq \left( \frac{L_2 - \alpha}{L_2 + \alpha} \right)^{2(j-1)} \mathbb{E} \| \Theta_t^0 - \Theta^*(\Psi_t^0) \|^2. \tag{15}$$

With the help of the above lemmas, we could give the estimation property of the $\frac{\partial L_{\text{global}}(\Psi_t^0, \Theta_t^{\tau})}{\partial \Psi_t^0}$ approximating $\nabla F(\Psi_t^0)$. The result is presented in the following Proposition 1.

**Proposition 1.** Under Assumptions 1-4, choose stepsize $\eta$ to be $\frac{2}{L_2+\alpha}$ and suppose $\alpha < L_2$, we have

$$\mathbb{E}\|\frac{\partial L_{\text{global}}(\Psi_t^0, \Theta_t^\tau; \mathcal{S}_0')}{\partial \Psi_t^0} - \nabla F(\Psi_t^0)\| \leq (L_2 + \frac{L_2^2}{\alpha})[\left(\frac{L_2 - \alpha}{L_2 + \alpha}\right)^\tau$$

$$\sqrt{\Delta}] + L_1[\frac{L_2(1 - \frac{2}{L_2+\alpha}\alpha)^\tau}{\alpha}$$

$$+ \frac{2}{L_2 + \alpha}(\frac{L_2 L_4}{\alpha} + L_3)\sqrt{\Delta} \tag{16}$$

$$\frac{(1 - \frac{2}{L_2+\alpha}\alpha)^\tau}{1 - \frac{2}{L_2+\alpha}\alpha - \frac{L_2-\alpha}{L_2+\alpha}}] + \frac{L_1}{\sqrt{N_b}}.$$

**Proof.** Using the triangle inequality, we have

$$\mathbb{E}\|\frac{\partial L_{\text{global}}(\Psi_t^0, \Theta_t^\tau; \mathcal{S}_0')}{\partial \Psi_t^0} - \nabla F(\Psi_t^0)\|$$

$$= \mathbb{E}\|\frac{\partial L_{\text{global}}(\Psi_t^0, \Theta_t^\tau; \mathcal{S}_0')}{\partial \Psi_t^0} - \frac{\partial \mathcal{L}_{\text{global}}(\Psi_t^0, \Theta_t^\tau)}{\partial \Psi_t^0}$$

$$+ \frac{\partial \mathcal{L}_{\text{global}}(\Psi_t^0, \Theta_t^\tau)}{\partial \Psi_t^0} - \nabla F(\Psi_t^0)\|$$

$$\leq \mathbb{E}\|\frac{\partial L_{\text{global}}(\Psi_t^0, \Theta_t^\tau; \mathcal{S}_0')}{\partial \Psi_t^0} - \frac{\partial \mathcal{L}_{\text{global}}(\Psi_t^0, \Theta_t^\tau)}{\partial \Psi_t^0}\| \tag{17}$$

$$+ \mathbb{E}\|\frac{\partial \mathcal{L}_{\text{global}}(\Psi_t^0, \Theta_t^\tau)}{\partial \Psi_t^0} - \nabla F(\Psi_t^0)\|$$

$$\overset{(i)}{\leq} \frac{L_1}{\sqrt{N_b}} + \mathbb{E}\|\frac{\partial \mathcal{L}_{\text{global}}(\Psi_t^0, \Theta_t^\tau)}{\partial \Psi_t^0} - \nabla F(\Psi_t^0)\|.$$

where (i) follows from equation 12.

Then, we need to give an upper bound for $\mathbb{E}\|\frac{\partial \mathcal{L}_{\text{global}}(\Psi_t^0, \Theta_t^\tau)}{\partial \Psi_t^0} - \nabla F(\Psi_t^0)\|$. Using

$$\nabla F(\Psi_t^0) = \nabla \mathcal{L}_{\text{global}}(\Psi_t^0, \Theta^*(\Psi_t^0)) + \frac{\partial \Theta^*(\Psi_t^0)}{\partial \Psi_t^0}\nabla_\Theta \mathcal{L}_{\text{global}}(\Psi_t^0, \Theta^*(\Psi_t^0))$$

and

$$\frac{\partial L_{\text{global}}(\Psi_t^0, \Theta_t^\tau)}{\partial \Psi_t^0} = \nabla_\Psi \mathcal{L}_{\text{global}}(\Psi_t^0, \Theta_t^\tau) + \frac{\partial \Theta_t^\tau}{\partial \Psi_t^0}\nabla_\Theta \mathcal{L}_{\text{global}}(\Psi_t^0, \Theta_t^\tau),$$

we have

$$\mathbb{E}\|\frac{\partial \mathcal{L}_{\text{global}}(\Psi_t^0, \Theta_t^\tau)}{\partial \Psi_t^0} - \nabla F(\Psi_t^0)\|$$

$$\leq L_2 \mathbb{E}\|\Theta_t^\tau - \Theta^*(\Psi_t^0)\| + L_1 \mathbb{E}\|\frac{\partial \Theta_t^\tau}{\partial \Psi_t^0} - \frac{\partial \Theta^*(\Psi_t^0)}{\partial \Psi_t^0}\| \tag{18}$$

$$+ L_2 \mathbb{E}(\|\frac{\partial \Theta^*(\Psi_t^0)}{\partial \Psi_t^0}\|\|\Theta_t^\tau - \Theta^*(\Psi_t^0)\|).$$

Now we want to bound $\mathbb{E}\|\frac{\partial \Theta_t^\tau}{\partial \Psi_t^0} - \frac{\partial \Theta^*(\Psi_t^0)}{\partial \Psi_t^0}\|$ first.

Recall the update method

$$\Theta_t^j = \Theta_t^{j-1} - \eta \nabla_\Theta \mathcal{L}_{\text{local}}(\Psi_t^0, \Theta_t^{j-1})$$

and use the chain rule on it, we have

$$\frac{\partial \Theta_t^j}{\partial \Psi_t^0} = \frac{\partial \Theta_t^{j-1}}{\partial \Psi_t^0} - \eta(\nabla_\Psi \nabla_\Theta \mathcal{L}_{\text{local}}(\Psi_t^0, \Theta_t^{j-1})$$

$$+ \frac{\partial \Theta_t^{j-1}}{\partial \Psi_t^0}\nabla_\Theta^2 \mathcal{L}_{\text{local}}(\Psi_t^0, \Theta_t^{j-1})). \tag{19}$$

For $\Theta^*(\Psi_t^0)$ is the optimal solution of $\mathcal{L}_{\text{local}}(\Psi_t^0, \Theta)$, we have $\nabla_\Theta \mathcal{L}_{\text{local}}(\Psi_t^0, \Theta^*(\Psi_t^0)) = 0$. Then, using the chain rule, we have

$$\nabla_\Psi \nabla_\Theta \mathcal{L}_{\text{local}}(\Psi_t^0, \Theta^*(\Psi_t^0)) + \frac{\partial \Theta^*(\Psi_t^0)}{\partial \Psi_t^0} \nabla_\Theta^2 \mathcal{L}_{\text{local}}(\Psi_t^0, \Theta^*(\Psi_t^0)) = 0. \tag{20}$$

Combining equation 19 and equation 20, the following equation holds

$$\begin{aligned}
\frac{\partial \Theta_t^j}{\partial \Psi_t^0} - \frac{\partial \Theta^*(\Psi_t^0)}{\partial \Psi_t^0} &= \frac{\partial \Theta_t^{j-1}}{\partial \Psi_t^0} - \frac{\partial \Theta^*(\Psi_t^0)}{\partial \Psi_t^0} \\
&\quad - \eta \left( \frac{\partial \Theta_t^{j-1}}{\partial \Psi_t^0} - \frac{\partial \Theta^*(\Psi_t^0)}{\partial \Psi_t^0} \right) \nabla_\Theta^2 \mathcal{L}_{\text{local}}(\Psi_t^0, \Theta_t^{j-1}) \\
&\quad - \eta \frac{\partial \Theta^*(\Psi_t^0)}{\partial \Psi_t^0} (\nabla_\Theta^2 \mathcal{L}_{\text{local}}(\Psi_t^0, \Theta_t^{j-1}) - \nabla_\Theta^2 \mathcal{L}_{\text{local}}(\Psi_t^0, \Theta^*(\Psi_t^0))).
\end{aligned} \tag{21}$$

Based on equation 20, we have

$$\left\| \frac{\partial \Theta^*(\Psi_t^0)}{\partial \Psi_t^0} \right\| \le \frac{L_2}{\alpha}. \tag{22}$$

With the help of Assumption 3, Lemma 2, equation 21 and equation 22, the following bound holds

$$\begin{aligned}
&\mathbb{E} \left\| \frac{\partial \Theta_t^j}{\partial \Psi_t^0} - \frac{\partial \Theta^*(\Psi_t^0)}{\partial \Psi_t^0} \right\| \\
&\le \mathbb{E}(\| I - \eta \nabla_\Theta^2 \mathcal{L}_{\text{local}}(\Psi_t^0, \Theta_t^{j-1}) \| \| \frac{\partial \Theta_t^{j-1}}{\partial \Psi_t^0} - \frac{\partial \Theta^*(\Psi_t^0)}{\partial \Psi_t^0} \|) \\
&\quad + \eta \frac{L_2}{\alpha} (\mathbb{E} \| \nabla_\Theta^2 \mathcal{L}_{\text{local}}(\Psi_t^0, \Theta_t^{j-1}) - \nabla_\Theta^2 \mathcal{L}_{\text{local}}(\Psi_t^0, \Theta^*(\Psi_t^0)) \| \\
&\quad + \eta \mathbb{E} \| \nabla_\Psi \nabla_\Theta \mathcal{L}_{\text{local}}(\Psi_t^0, \Theta_t^{j-1}) - \nabla_\Psi \nabla_\Theta \mathcal{L}_{\text{local}}(\Psi_t^0, \Theta_t^*(\Psi_t^0)) \| \\
&\le (1 - \eta\alpha) \mathbb{E} \left\| \frac{\partial \Theta_t^{j-1}}{\partial \Psi_t^0} - \frac{\partial \Theta^*(\Psi_t^0)}{\partial \Psi_t^0} \right\| + \eta (\frac{L_2 L_4}{\alpha} + L_3) \mathbb{E} \| \Theta_t^{j-1} - \Theta^*(\Psi_t^0) \|.
\end{aligned} \tag{23}$$

For the choice of $\eta = \frac{2}{L_2 + \alpha}$, Lemma 4 holds. With the result in equation 15 and Assumption 4, we have

$$\begin{aligned}
\mathbb{E} \| \Theta_t^{j-1} - \Theta^*(\Psi_t^0) \| &\le \left( \frac{L_2 - \alpha}{L_2 + \alpha} \right)^{j-1} \sqrt{\mathbb{E} \| \Theta_t^0 - \Theta^*(\Psi_t^0) \|} \\
&\le \left( \frac{L_2 - \alpha}{L_2 + \alpha} \right)^{j-1} \sqrt{\Delta}.
\end{aligned} \tag{24}$$

Telescoping equation 23 over $j$ from 0 to $\tau$ and combining the result with equation 24 yields

$$\begin{aligned}
\mathbb{E} \left\| \frac{\partial \Theta_t^\tau}{\partial \Psi_t^0} - \frac{\partial \Theta^*(\Psi_t^0)}{\partial \Psi_t^0} \right\| &\le (1 - \eta\alpha)^\tau \mathbb{E} \left\| \frac{\partial \Theta_t^0}{\partial \Psi_t^0} - \frac{\partial \Theta^*(\Psi_t^0)}{\partial \Psi_t^0} \right\| \\
&\quad + \eta \left( \frac{L_2 L_4}{\alpha} + L_3 \right) \sum_{j=0}^{\tau-1} (1 - \eta\alpha)^{\tau-1-j} \\
&\quad \left[ \left( \frac{L_2 - \alpha}{L_2 + \alpha} \right)^j \sqrt{\Delta} \right] \\
&\le \frac{L_2 (1 - \eta\alpha)^\tau}{\alpha} \\
&\quad + \eta (\frac{L_2 L_4}{\alpha} + L_3) \sqrt{\Delta} \frac{(1 - \eta\alpha)^\tau}{1 - \eta\alpha - \frac{L_2 - \alpha}{L_2 + \alpha}}.
\end{aligned} \tag{25}$$

Take expedition of both sides of equation 18, plugging equation 22, equation 24 and equation 25 into it yields

$$\mathbb{E}\|\frac{\partial \mathcal{L}_{\text{global}}(\Psi_t^0, \Theta_t^\tau)}{\partial \Psi_t^0} - \nabla F(\Psi_t^0)\| \leq (L_2 + \frac{L_2^2}{\alpha})[\left(\frac{L_2 - \alpha}{L_2 + \alpha}\right)^\tau \sqrt{\Delta}]$$
$$+ L_1[\frac{L_2(1 - \frac{2}{L_2+\alpha}\alpha)^\tau}{\alpha}$$
$$+ \frac{2}{L_2 + \alpha}(\frac{L_2 L_4}{\alpha} + L_3)\sqrt{\Delta}$$
$$\frac{(1 - \frac{2}{L_2+\alpha}\alpha)^\tau}{1 - \frac{2}{L_2+\alpha}\alpha - \frac{L_2-\alpha}{L_2+\alpha}}]. \tag{26}$$

Finally, plugging equation 26 into equation 17 yields equation 16. Thus, we complete the proof of Propostion 1. $\square$

To deal with multiple updates of the active party with a fixed $t$, we need to use a technology called virtual updates (Yang et al., 2022). Specifically, we introduce a virtual parameter $\Theta_{t,j'}^\tau$ which could be obtained by local updates when $\Psi_t^{j'}$ is given. With the help of Proposition 1, we could obtain the bound for $\mathbb{E}\|\frac{\partial L_{\text{global}}(\Psi_t^{j'}, \Theta_{t,j'}^\tau; \mathcal{S}'_{j'})}{\partial \Psi_t^{j'}} - \nabla F(\Psi_t^{j'})\|$. Now, we want to give a bound for $\mathbb{E}\|\frac{\partial L_{\text{global}}(\Psi_t^{j'}, \Theta_t^\tau; \mathcal{S}'_{j'})}{\partial \Psi_t^{j'}} - \nabla F(\Psi_t^{j'})\|$. The result is as follows

**Proposition 2.** Following the conditions in Proposition 1, we have

$$\mathbb{E}\|\frac{\partial L_{\text{global}}(\Psi_t^{j'}, \Theta_t^\tau; \mathcal{S}'_{j'})}{\partial \Psi_t^{j'}} - \nabla F(\Psi_t^{j'})\| \leq (L_2 + \frac{L_2^2}{\alpha})[\left(\frac{L_2 - \alpha}{L_2 + \alpha}\right)^\tau$$
$$\sqrt{\Delta}] + L_1[\frac{L_2(1 - \frac{2}{L_2+\alpha}\alpha)^\tau}{\alpha}$$
$$+ \frac{2}{L_2 + \alpha}(\frac{L_2 L_4}{\alpha} + L_3)\sqrt{\Delta}$$
$$\frac{(1 - \frac{2}{L_2+\alpha}\alpha)^\tau}{1 - \frac{2}{L_2+\alpha}\alpha - \frac{L_2-\alpha}{L_2+\alpha}}] + \frac{L_1}{\sqrt{N_b}} + L_2\Delta. \tag{27}$$

**Proof.** Using the triangle inequality, we have

$$\|\frac{\partial L_{\text{global}}(\Psi_t^{j'}, \Theta_t^\tau; \mathcal{S}'_{j'})}{\partial \Psi_t^{j'}} - \nabla F(\Psi_t^{j'})\|$$
$$\leq \|\frac{\partial L_{\text{global}}(\Psi_t^{j'}, \Theta_t^\tau; \mathcal{S}'_{j'})}{\partial \Psi_t^{j'}} - \frac{\partial L_{\text{global}}(\Psi_t^{j'}, \Theta_{t,j'}^\tau; \mathcal{S}'_{j'})}{\partial \Psi_t^{j'}}\| \tag{28}$$
$$+ \|\frac{\partial L_{\text{global}}(\Psi_t^{j'}, \Theta_{t,j'}^\tau; \mathcal{S}'_{j'})}{\partial \Psi_t^{j'}} - \nabla F(\Psi_t^{j'})\|$$

Take expectation of both sides of equation 28, with the help of Assumption 2 and 5, the following result holds

$$\mathbb{E}\|\frac{\partial L_{\text{global}}(\Psi_t^{j'}, \Theta_t^{\tau}; \mathcal{S}_{j'}')}{\partial \Psi_t^{j'}} - \nabla F(\Psi_t^{j'})\|$$

$$\leq \mathbb{E}\|\frac{\partial L_{\text{global}}(\Psi_t^{j'}, \Theta_t^{\tau}; \mathcal{S}_{j'}')}{\partial \Psi_t^{j'}} - \frac{\partial L_{\text{global}}(\Psi_t^{j'}, \Theta_{t,j'}^{\tau}; \mathcal{S}_{j'}')}{\partial \Psi_t^{j'}}\|$$

$$+ \mathbb{E}\|\frac{\partial L_{\text{global}}(\Psi_t^{j'}, \Theta_{t,j'}^{\tau}; \mathcal{S}_{j'}')}{\partial \Psi_t^{j'}} - \nabla F(\Psi_t^{j'})\| \tag{29}$$

$$\leq L_2 \|\Theta_t^{\tau} - \Theta_{t,j'}^{\tau}\| + \mathbb{E}\|\frac{\partial L_{\text{global}}(\Psi_t^{j'}, \Theta_{t,j'}^{\tau}; \mathcal{S}_{j'}')}{\partial \Psi_t^{j'}} - \nabla F(\Psi_t^{j'})\|$$

$$\leq L_2 \Delta + \mathbb{E}\|\frac{\partial L_{\text{global}}(\Psi_t^{j'}, \Theta_{t,j'}^{\tau}; \mathcal{S}_{j'}')}{\partial \Psi_t^{j'}} - \nabla F(\Psi_t^{j'})\|.$$

Plugging in the result in Proposition 1 into equation 29 yields equation 27. Thus, we complete the proof of Propostion 2. □

**Proof for Theorem 1.** Based on the $L_0$-smoothness of $F(\Psi)$ established in Lemma 3, we have

$$F(\Psi_t^{j'+1}) \leq F(\Psi_t^{j'}) + \langle \nabla F(\Psi_t^{j'}), \Psi_t^{j'+1} - \Psi_t^{j'} \rangle$$

$$+ \frac{L_0}{2}\|\Psi_t^{j'+1} - \Psi_t^{j'}\|^2$$

$$\leq F(\Psi_t^{j'}) - \eta'\langle \nabla F(\Psi_t^{j'}), \frac{\partial \mathcal{L}_{\text{global}}(\Psi_t^{j'}, \Theta_t^{\tau})}{\partial \Psi_t^{j'}} - \nabla F(\Psi_t^{j'})\rangle$$

$$- \eta'\|\nabla F(\Psi_t^{j'})\|^2 + \eta'^2 L_0\|\nabla F(\Psi_t^{j'})\|^2 \tag{30}$$

$$+ \eta'^2 L_0\|\frac{\partial \mathcal{L}_{\text{global}}(\Psi_t^{j'}, \Theta_t^{\tau})}{\partial \Psi_t^{j'}} - \nabla F(\Psi_t^{j'})\|^2$$

$$\leq F((\Psi_t^{j'}) - (\frac{\eta'}{2} - \eta'^2 L_0)\|\nabla F((\Psi_t^{j'})\|^2$$

$$+ (\frac{\eta'}{2} + \eta'^2 L_0)\|\frac{\partial \mathcal{L}_{\text{global}}(\Psi_t^{j'}, \Theta_t^{\tau})}{\partial \Psi_t^{j'}} - \nabla F(\Psi_t^{j'})\|^2.$$

Telescoping equation 30 over $j'$ from 0 to $\tau' - 1$ and taking expection of both sides yields

$$\mathbb{E}F(\Psi_t^{\tau'}) \leq \mathbb{E}F(\Psi_t^0) - (\frac{\eta'}{2} - \eta'^2 L_0)\mathbb{E}\sum_{j'=0}^{\tau'-1}\|\nabla F(\Psi_t^{j'})\|^2$$

$$+ (\eta' + 2\eta'^2 L_0)\tau'\{(L_2 + \frac{L_2^2}{\alpha})[\left(\frac{L_2 - \alpha}{L_2 + \alpha}\right)^{\tau}$$

$$\sqrt{\Delta}] + L_1[\frac{L_2(1 - \frac{2}{L_2+\alpha}\alpha)^{\tau}}{\alpha} \tag{31}$$

$$+ \frac{2}{L_2 + \alpha}(\frac{L_2 L_4}{\alpha} + L_3)\sqrt{\Delta}$$

$$\frac{(1 - \frac{2}{L_2+\alpha}\alpha)^{\tau}}{1 - \frac{2}{L_2+\alpha}\alpha - \frac{L_2-\alpha}{L_2+\alpha}}] + \frac{L_1}{\sqrt{N_b}}\}^2$$

$$+ (\eta' + 2\eta'^2 L_0)L_2^2\Delta^2(\tau' - 1).$$

Telescoping equation 31 over $t$ from 0 to $T-1$ yields equation 11, we complete the proof of Theorem 1. □

