# OpenReview forum: "BoMM: Multi-Modality Large-Small Model Bidirectional Collaboration"
_ICLR.cc/2026/Conference — Submitted to ICLR 2026_

### Official Review · Reviewer_pFUw · 2025-10-27

**Soundness:** 2
**Presentation:** 2
**Contribution:** 2
**Rating:** 4
**Confidence:** 3

**Summary:**

This paper introduces a novel framework, BoMM, for bidirectional collaboration between a Multi-Modality Large Model (MM-LM) and multiple parties' local small models (SMs). It aims to address two key challenges: 1) Modality Alignment Scarcity, and 2) Modality Completeness Gap. BoMM proposes a two-part solution: 1) a Global Prototype-Guided Alignment (GPGA) strategy is used to mitigate alignment scarcity; and 2) a Preference-Driven Modality Adaptive Completion (PDMAC) method is designed to handle the modality completeness gap. Inspired by Direct Preference Optimization (DPO), it trains the MM-LM's generator to complete missing modalities for the SMs, using global prototypes to form preference pairs without requiring labels. Extensive experiments on MS COCO and CUB-200-2011 datasets demonstrate a performance improvement over state-of-the-art baselines.

**Strengths:**

1. The proposed BoMM directly targets the identified challenges. GPGA and PDMAC are designed to bootstrap each other. The pipeline and figure explain this design clearly.
2. Adapting DPO preference loss to guide the MM-LM generator toward prototype-anchored preferred completions is a sensible way to keep generation personalized.
3. Evaluations across multiple scenarios show the robustness and effectiveness of the proposed method.

**Weaknesses:**

1. Since this paper focuses on the settings of large-small model collaboration, it is recommended to elaborate more on the practical values of this setting in the Introduction. Additionally, organize the related work around studies addressing the same setting to introduce the literature, rather than general techniques like meta pseudo labels.
2. The notation may need further clarification. For example:
- What do $x$, $y$ represent in Section 3.1?
- If $y$ denotes the label, then how is it defined for "unlabeled samples"?
- How are multimodal data samples represented in the notation $(x_i, y_i)\in D_k$?
- From Section 3.1, the aligned samples are defined as those with the same ID across parties, while the unaligned ones are those with sample IDs found only in the party. However, in Figure 2, both Party A and Party B show samples with ID = 1, yet the contents differ. Should these be considered aligned?
- Symbols like $\delta\mathcal{H}$ are not defined.
- $\Delta$ is reused for different meanings.
3. The convergence proof relies on Assumptions 1-4. These can be hard to justify for deep neural networks, which are usually non-convex and not strongly convex. It is necessary to discuss how these assumptions map to the practical setup. Please clarify the gap and provide empirical evidence.
4. Additional analysis is necessary, including the computational complexity, scalability, and privacy/leakage risks.
5. Some typos like "Rp eedforward" in Line 263 need to be fixed.
6. In the Introduction, this paper identifies a gap as existing completion approaches struggle with severe or even complete modality-missing cases. Can the proposed method address this gap? Experiments seem to evaluate only mild missing rates (0.1 to 0.5).
7. The identified main challenges, i.e., modality alignment and modality completeness, are very common and well-studied in multimodal learning. How do the proposed solutions compare to classic and representative approaches in the literature? It is suggested to provide such discussions and empirical evaluations.

**Questions:**

1. Since this paper focuses on the settings of large-small model collaboration, it is recommended to elaborate more on the practical values of this setting in the Introduction. Additionally, organize the related work around studies addressing the same setting to introduce the literature, rather than general techniques like meta pseudo labels.
2. The notation may need further clarification. For example:
- What do $x$, $y$ represent in Section 3.1?
- If $y$ denotes the label, then how is it defined for "unlabeled samples"?
- How are multimodal data samples represented in the notation $(x_i, y_i)\in D_k$?
- From Section 3.1, the aligned samples are defined as those with the same ID across parties, while the unaligned ones are those with sample IDs found only in the party. However, in Figure 2, both Party A and Party B show samples with ID = 1, yet the contents differ. Should these be considered aligned?
- Symbols like $\delta\mathcal{H}$ are not defined.
- $\Delta$ is reused for different meanings.
3. The convergence proof relies on Assumptions 1-4. These can be hard to justify for deep neural networks, which are usually non-convex and not strongly convex. It is necessary to discuss how these assumptions map to the practical setup. Please clarify the gap and provide empirical evidence.
4. Additional analysis is necessary, including the computational complexity, scalability, and privacy/leakage risks.
5. Some typos like "Rp eedforward" in Line 263 need to be fixed.
6. In the Introduction, this paper identifies a gap as existing completion approaches struggle with severe or even complete modality-missing cases. Can the proposed method address this gap? Experiments seem to evaluate only mild missing rates (0.1 to 0.5).
7. The identified main challenges, i.e., modality alignment and modality completeness, are very common and well-studied in multimodal learning. How do the proposed solutions compare to classic and representative approaches in the literature? It is suggested to provide such discussions and empirical evaluations.

---

### Official Review · Reviewer_vhdw · 2025-10-31

**Soundness:** 2
**Presentation:** 2
**Contribution:** 2
**Rating:** 2
**Confidence:** 4

**Summary:**

The paper explores a framework where a cloud-side multi-modal large model collaborates with smaller domain-specific models to improve each other’s performance. The proposed BoMM framework introduces a global prototype-guided alignment method to identify aligned cross-party samples and a preference-driven modality adaptive completion method to fill in missing modalities dynamically. Experiments on multi-modal datasets show consistent gains over existing methods.

**Strengths:**

1. The paper addresses a relatively under-explored setting, multi-modality large–small model collaboration, and clearly articulates the motivation behind studying it.

2. The method appears compatible with existing multi-modal models, suggesting potential extensibility to other settings or architectures.

3. The paper provides both empirical and theoretical components, which adds completeness to the overall presentation.

**Weaknesses:**

1. The motivation is conceptually interesting but practically hollow. While the paper claims to target realistic multi-modality large–small collaboration scenarios, it never provides a convincing real-world use case or system setup. As a result, the motivation feels largely hypothetical. The paper frames a federated-like problem but does not reproduce its practical constraints or challenges (e.g., non-IID data, network overhead, privacy leakage).


2. The so-called “DPO-inspired modality adaptive completion” is essentially a heuristic pairwise preference loss where the “preferred” sample is chosen from global prototypes and the “non-preferred” one is the generator’s own output.
It borrows DPO’s pairwise logistic form but lacks human preference data, reference policy comparison, and proper reward modeling, making it more of a prototype-guided contrastive loss than true direct preference optimization.

3. The proposed Dynamic Sample Scheduler in Section 4.1.2 is mathematically over-engineered yet conceptually simple. It merely ranks generated samples by heuristic “reward” scores derived from loss changes. Despite its elaborate formulation, it functions as an ad-hoc filtering mechanism without theoretical grounding. This type of threshold based methods are commonly seen in machine learning community.

4. The complexity and convergence analysis in section 4.3 is very generical and meaningless. $O(1-\sqrt{T})$ is not specific to BoMM; it’s the typical nonconvex stochastic rate under smoothness assumptions, and it’s common in bi-level analyses built on similar lemmas (smoothness of $F$, tracking bounds, etc.). I checked the proof in appendix. The proof does not analyze the “prototype-guided alignment” or the “preference-driven completion” mechanics per se; it abstracts them away into generic losses and smoothness constants. So it doesn’t tell you that GPGA or the scheduler/DPO-like module is stable or optimal, only that the outer bi-level loop can converge under regularity. I fully understand that a through analysis for nonconvex network is still an open problem in ML theory. But I don't think it is ok claim this analysis as an important contribution of your work.

5. Baseline selection is incomplete and partly unfair. The baselines (LOCAL, MM-LOCAL, Vanilla, MM-AC, MM-UC) are mostly paper-internal variants rather than strong external comparisons.  No comparison is made against:

* existing multimodal federated learning methods (e.g., FedMBridge[1], FedMKT[2], FedMVP [3]);

* recent missing-modality completion or multimodal alignment works using diffusion or contrastive generation (e.g., VLMo[4]).


[1] Chen, J. &amp; Zhang, A.. (2024). FedMBridge: Bridgeable Multimodal Federated Learning. Proceedings of the 41st International Conference on Machine Learning, in Proceedings of Machine Learning Research 235:7667-7686 Available from https://proceedings.mlr.press/v235/chen24ba.html.

[2] Tao Fan, Guoqiang Ma, Yan Kang, Hanlin Gu, Yuanfeng Song, Lixin Fan, Kai Chen, and Qiang Yang. 2025. FedMKT: Federated Mutual Knowledge Transfer for Large and Small Language Models. In Proceedings of the 31st International Conference on Computational Linguistics, pages 243–255, Abu Dhabi, UAE. Association for Computational Linguistics.

[3] Che, L., Wang, J., Liu, X., Ma, F.(2024). Leveraging Foundation Models for Multi-modal Federated Learning with Incomplete Modality. ECML-PKDD 2024

[4] Hangbo Bao, Wenhui Wang, Li Dong, Qiang Liu, Owais Khan Mohammed, Kriti Aggarwal, Subhojit Som, Songhao Piao, and Furu Wei. 2022. VLMo: unified vision-language pre-training with mixture-of-modality-experts. In Proceedings of the 36th International Conference on Neural Information Processing Systems (NIPS '22). Curran Associates Inc., Red Hook, NY, USA, Article 2384, 32897–32912.

**Questions:**

See Weaknesses.

---

### Official Review · Reviewer_vJg4 · 2025-11-01

**Soundness:** 3
**Presentation:** 3
**Contribution:** 2
**Rating:** 6
**Confidence:** 2

**Summary:**

This paper proposes a framework called BoMM (Bidirectional Multi-Modality Large-Small Model Collaboration), which aims to address two major challenges commonly encountered in multi-modal collaborative learning: modality alignment scarcity and modality completeness gap. The framework is built upon two core mechanisms. The first is the Global Prototype-Guided Alignment (GPGA) strategy, which leverages optimal transport to compute similarity distributions between unaligned samples and global prototypes, enabling the identification of potential cross-party aligned samples even under zero-alignment conditions, thereby facilitating knowledge transfer from small models to the large model. The second is the Preference-Driven Modality Adaptive Completion (PDMAC) method, which integrates Direct Preference Optimization (DPO) with a dynamic sample scheduling mechanism, allowing the multi-modal large model to generate domain-specific missing modalities and achieve reverse knowledge transfer from the large model to the small models. Comprehensive experiments are conducted on the MS COCO and CUB-200-2011 datasets under various modality-missing rates and alignment ratios. The results show that BoMM achieves significant performance improvements over existing state-of-the-art methods on both cloud and local models, with a maximum gain of 6.64%, and demonstrates superior generation quality as measured by CLIPScore. The authors further validate the effectiveness of each component through ablation studies and hyperparameter sensitivity analysis. Overall, this work represents the first systematic exploration of bidirectional collaboration between large and small models in multi-modal settings, providing a novel theoretical and practical framework for multi-modal learning, federated collaborative training, and privacy-preserving cross-modal knowledge sharing, with substantial research significance and potential practical value.

**Strengths:**

- This paper systematically introduces a novel research paradigm of bidirectional collaboration between large and small models in multi-modal settings, breaking through the traditional limitation of unidirectional distillation or single-modality cooperation. The problem formulation is forward-looking and demonstrates strong theoretical originality.
- The proposed Global Prototype-Guided Alignment (GPGA) strategy leverages optimal transport and prototype similarity distributions to achieve unsupervised cross-party sample alignment. Even under zero-alignment conditions, it can effectively identify potential matching samples, thereby alleviating the problem of modality alignment scarcity.
- The paper is well-written with clear logical structure and detailed illustrative figures. In the experimental section, the effectiveness of each proposed component is thoroughly validated through extensive visualizations and ablation studies from multiple perspectives.
- The Preference-Driven Modality Adaptive Completion (PDMAC) method integrates Direct Preference Optimization (DPO) with a dynamic sample scheduling mechanism, enabling the multi-modal large model to adaptively generate missing modalities according to domain-specific characteristics and achieve reverse knowledge transfer from large to small models.
- The two modules work synergistically to form a stable and efficient “alignment–completion” closed-loop optimization mechanism. The overall approach is logically coherent, structurally sound, and exhibits strong systematic design and scalability.

**Weaknesses:**

All experiments are conducted on standard open datasets rather than in real-world distributed or privacy-constrained environments, leaving the scalability and robustness of BoMM under practical federated or heterogeneous settings unverified. Moreover, the paper does not evaluate the stability of BoMM when deployed across heterogeneous model architectures (e.g., Vision Transformers vs. CNNs) or under non-IID data distributions.
- The computational and communication overhead of the proposed method is not sufficiently quantified, and no comparison is provided regarding training efficiency or resource consumption relative to other collaborative multi-modal frameworks.
- The convergence analysis largely follows a standard bilevel optimization formulation without providing module-specific theoretical modeling for GPGA or DPO, resulting in a weak connection between the theoretical derivation and the actual algorithmic mechanisms.
- The paper does not directly compare BoMM with recent multi-modal collaborative learning frameworks such as FedMM, OpenFedLLM, or Multi-FL, relying instead on simple local or vanilla aggregation baselines, which may underestimate the competitiveness of existing methods.
- The sensitivity of parameters β and λ in the PDMAC reward function is not discussed, which may affect the diversity and stability of the generated modalities. Furthermore, the potential for false alignment in GPGA under zero-alignment initialization remains an open concern requiring empirical validation.

**Questions:**

See weaknesses.

---

### Official Review · Reviewer_FtV8 · 2025-11-07

**Soundness:** 3
**Presentation:** 3
**Contribution:** 3
**Rating:** 6
**Confidence:** 4

**Summary:**

This paper introduces BoMM, a framework for multi-modality large-small model (MM-LM & SM) bidirectional collaboration, addressing two challenges in distributed multimodal learning: Modality alignment scarcity and Modality completeness gap, missing modalities in each party’s dataset due to device or annotation limitations. BoMM proposes Global Prototype-Guided Alignment (GPGA), identifies aligned samples across parties using optimal transport-based similarity distributions between unaligned samples and global prototypes, and Preference-Driven Modality Adaptive Completion (PDMAC), integrates direct preference optimization with dynamic sample scheduling to generate missing modalities conditioned on global prototypes.
Theoretical analysis shows O(1/\sqrt{T}) convergence under standard bilevel optimization assumptions. Experiments on MS COCO and CUB-200-2011 under multiple missing-modality scenarios demonstrate up to 6.64% improvement.

--soundness--
The proposed methodology is mathematically and algorithmically consistent, combining established elements (DPO, OT-based alignment, bilevel optimization). The convergence proof is coherent enough. But there are still some problems:
a)	The experimental comparison lacks stronger multimodal LLM baselines (e.g., locally fine-tuned LLAMA-1.5-7B, for it is not too large to fine-tune). I am curious if BoMM could catch up those well-aligned MM-LMs.
b)	The PDMAC reward definition (Eq. 5) mixes terms with unclear normalization; the interaction between sample- and batch-level components is not theoretically justified.
c)	All ablation studies were about PDMAC. No ablation studies were focused on GPGA.
--presentation--
The paper is generally written and structured, though some (e.g., “privacy-preserving by exchanging intermediate representations”) are mentioned but not analyzed. Also, Mathematical notation is inconsistent in several places: x^{v,t}_{i,k}, g^p_i/g^q_i, and reward function terms switched notation in sections.
--contribution--
The paper’s primary novelty lies in integrating large-small model collaboration with multimodal incompleteness and unalignment, two aspects rarely tackled together. Its bidirectional knowledge transfer between SMs and MM-LM also marks a notable conceptual advance over unidirectional distillation.
However, the proposed modules are merely adaptations of established concepts (prototype alignment, preference optimization) rather than entirely new algorithms. Additionally, the evaluation scope, covering only two datasets and synthetic missing-modality simulations, is narrow for a “large-model” assertion.

**Strengths:**

a)	Novel problem formulation: first attempt to unify multimodal incompleteness and cross-party alignment within a large-small collaborative framework.
b)	Solid methodological grounding: integrates DPO, prototype-based alignment, and bilevel optimization coherently.
c)	Reasonable theoretical analysis: convergence at O(1/\sqrt{T}) is okay but demonstrates technical rigor.
d)	Empirical evidence: consistent performance improvement across multiple scenarios and clear ablation validation of components.

**Weaknesses:**

a)	GPGA and PDMAC only extend existing methods (OT-based alignment, DPO) with little algorithmic innovation.
b)	Missing-modality settings are synthetic; no real-world multimodal missingness is assessed.
c)	The “cloud-side MM-LM” uses LLaVA-1.5-7B, but no comparison to other multimodal LLMs is provided. Also, it is not strong enough to be “cloud-side”.
d)	The paper claims privacy benefits (“exchange of intermediate representations”) but provides no empirical or theoretical justification, nor a privacy–utility tradeoff analysis.
e)	Some definitions (e.g., of global prototypes, modality sets, losses) appear fragmented across sections, requiring frequent back-and-forth reading.

**Questions:**

a) How sensitive is the system to reward scaling parameters (β, λ, μ)? Only μ is covered in the ablation.
b) Could the authors clarify if modality completion quality (via CLIPScore) correlates with downstream accuracy?
c) Is the dynamic scheduler differentiable and jointly trained with the generator, or is it post-hoc sample selection?
d) Please elaborate on whether privacy preservation is empirically measured or merely conceptual.

---

### Meta-Review · Area_Chair_44oD · 2026-01-10

**Summary:**

Reviewer scores are 6, 6, 2, and 4 (average 4.5). The main weaknesses center on:
- limited algorithmic novelty,
- insufficient experimental evaluation, including narrow evaluation (synthetic missingness, two datasets), incomplete baselines (omitting stronger MM-LMs and federated multimodal methods), generic convergence analysis, unclear reward normalization and sensitivity for β,λ,μ, and missing GPGA ablations,
- poor representation, including notation inconsistencies, and unsubstantiated privacy claims.

**Reviewer Concerns:**

No author rebuttal is provided, and these concerns remain.

**Reviewer Scores:**

As these concerns remain, all reviewers are likely to keep their rating unchanged.

---

### Decision · Program_Chairs · 2026-01-26

Reject